# Does COVID-19 Vaccination Cause Storage Lower Urinary Tract Symptoms?

**DOI:** 10.3390/jcm11102736

**Published:** 2022-05-12

**Authors:** Yu-Chen Chen, Yin-Chi Liang, Shuo-Jung Ho, Hao-Wei Chen, Yung-Shun Juan, Wei-Chung Tsai, Shu-Pin Huang, Jung-Ting Lee, Yu-Peng Liu, Chung-Yao Kao, Yen-Ko Lin, Cheng-Yu Long, Meng-Ni Wu, Chao-Ju Chen, Wen-Jeng Wu

**Affiliations:** 1Graduate Institute of Clinical Medicine, College of Medicine, Kaohsiung Medical University, Kaohsiung 80708, Taiwan; jennis7995@hotmail.com (Y.-C.C.); chanhoward21@hotmail.com (H.-W.C.); juanuro@gmail.com (Y.-S.J.); shpihu73@gmail.com (S.-P.H.); ypliu@kmu.edu.tw (Y.-P.L.); urolong@yahoo.com.tw (C.-Y.L.); berkeley0701@gmail.com (M.-N.W.); chaoju.chen@gmail.com (C.-J.C.); 2Department of Urology, Kaohsiung Medical University Hospital, Kaohsiung Medical University, Kaohsiung 80708, Taiwan; 3School of Medicine, Kaohsiung Medical University, Kaohsiung 80708, Taiwan; skychaser17@gmail.com (Y.-C.L.); u105001120@gap.kmu.edu.tw (S.-J.H.); 4Department of Urology, Kaohsiung Municipal Ta-Tung Hospital, Kaohsiung 80145, Taiwan; 5Division of Cardiology, Department of Internal Medicine, Kaohsiung Medical University Hospital, Kaohsiung 807377, Taiwan; u8501091@gap.kmu.edu.tw; 6Drug Development and Value Creation Research Center, Kaohsiung Medical University, Kaohsiung 80708, Taiwan; 7Si Wan College, National Sun-Yat Sen University, Kaohsiung 80424, Taiwan; celeste@g-mail.nsysu.edu.tw; 8Department of Electrical Engineering, National Sun-Yat Sen University, Kaohsiung 80424, Taiwan; cykao@mail.ee.nsysu.edu.tw; 9Division of Trauma and Surgical Critical Care, Department of Surgery, Kaohsiung Medical University Hospital, Kaohsiung Medical University, Kaohsiung 80708, Taiwan; yenko@ms16.hinet.net; 10Department of Medical Humanities and Education, College of Medicine, Kaohsiung Medical University, Kaohsiung 80708, Taiwan; 11Department of Obstetrics and Gynecology, Kaohsiung Medical University Hospital, Kaohsiung Medical University, Kaohsiung 80708, Taiwan

**Keywords:** COVID-19, lower urinary tract symptoms, vaccines, adverse effects

## Abstract

We investigated the storage lower urinary tract symptoms (LUTS) before and after the first dose of coronavirus disease 2019 (COVID-19) vaccine and the association between pre-vaccinated overactive bladder (OAB) and the worsening of storage LUTS following COVID-19 vaccination. This cross-sectional study in a third-level hospital in Taiwan used the validated pre- and post-vaccinated Overactive Bladder Symptom Score (OABSS). Diagnosis of OAB was made using pre-vaccinated OABSS. The deterioration of storage LUTS was assessed as the increased score of OABSS following vaccination. Of 889 subjects, up to 13.4% experienced worsened storage LUTS after vaccination. OAB was significantly associated with an increased risk of worsening urinary urgency (*p* = 0.030), frequency (*p* = 0.027), and seeking medical assistance due to urinary adverse events (*p* < 0.001) after vaccination. The OAB group faced significantly greater changes in OABSS-urgency (*p* = 0.003), OABSS-frequency (*p* = 0.025), and total OABSS (*p* = 0.014) after vaccination compared to those observed in the non-OAB group. Multivariate regression revealed that pre-vaccinated OAB (*p* = 0.003) was a risk for the deterioration of storage LUTS. In conclusion, storage LUTS may deteriorate after vaccination. OAB was significantly associated with higher risk and greater changes in worsening storage LUTS. Storage LUTS should be closely monitored after COVID-19 vaccination, especially in those OAB patients.

## 1. Introduction

The outbreak of severe acute respiratory syndrome coronavirus-2 (SARS-CoV-2) rapidly resulted in a pandemic and posed challenges for a successful vaccine outcome [1]. The US Food and Drug Administration has authorized mRNA vaccines to prevent COVID-19, including the BNT162b2 mRNA coronavirus disease 2019 (COVID-19) vaccine (Pfizer-BioNTech) and the mRNA-1273 COVID-19 vaccine (Moderna) in December 2020. Additionally, the Oxford University/AstraZeneca simian adenovirus-vectored AZD1222 vaccine has come online since December 2020. In Taiwan, Medigen Vaccine Biologics Corporation developed the MVC COVID-19 vaccine, which obtained Emergency Use Authorization approval from the Taiwanese government on 19 July 2021.

Regarding the post-vaccination adverse effects (AEs), several recent studies on self-reported side effects with the mRNA-based (Moderna and Pfizer-BioNTech) and viral vector-based (AstraZeneca) vaccine in healthcare sector workers have reported a broad spectrum of symptomatology, with the most common being localized symptoms, such as pain around injection site or a sore arm, and generalized symptoms, such as weakness or fatigue [1,2,3,4,5]. Among them, urological AEs were reported in three studies, in which the majority of the observed urological AEs were storage lower urinary tract symptoms (LUTS), including urinary urgency (occurrence rate: 0.75%–1.16%) and urinary frequency (occurrence rate: 0.37%) [1,5].

However, these three studies failed to evaluate the subjects’ baseline status of LUTS before vaccination, making it difficult to determine the true side effects of vaccination. In addition, these studies used only yes/no questions instead of validated urological questionnaires to evaluate the urinary AEs, which made it impossible to investigate the severity of the symptoms and might not truly reveal the urinary status, especially in those who did not know the accurate definition of the storage LUTS. To our knowledge, there have not been any reports on urological AEs following COVID-19 vaccination that focus specifically on a detailed change in LUTS before and after vaccination using validated urological questionnaires. In addition, interestingly, these reported urological AEs appear to be similar to the symptoms of overactive bladder (OAB). We further hypothesized that the worsening of the storage LUTS due to vaccination may be different between subjects with and without OAB. Therefore, the aim of this study was to provide an in-depth urological assessment of the storage LUTS before and after COVID-19 vaccination, as well as to evaluate the association between the OAB and the deterioration in the storage LUTS following COVID-19 vaccination.

## 2. Materials and Methods

### 2.1. Ethics Statement

The study protocol was approved by the institutional review board (IRB) of Kaohsiung Medical University Hospital (IRB No. KMUHIRB-E(I)-20210230).

### 2.2. Design and Sample Selection

A cross-sectional research was performed by an online survey questionnaire through an internet-based survey platform (Survey Cake), gathering anonymous responses during the early phase of COVID-19 vaccination. No personal identification was obtained. The Survey Cake web link was distributed to subjects via social media or face to face when they attended the tertiary hospital for COVID-19 vaccination. Informed consent was obtained at the beginning of the survey. The study inclusion criteria were age ≥ 20 years and the ability to complete the serial questionnaires. Subjects who voluntarily agreed and consented to proceed and who have received one of the COVID-19 vaccines approved in Taiwan (including two mRNA-based: Moderna and Pfizer-BioNTech; viral vector-based: AstraZeneca vaccine; and a recombinant protein vaccine, MVC-COV1901 [6]) were automatically allowed to move forward to answer following questions about the assessment of LUTS and the AEs after vaccination.

### 2.3. Duration of Study

The Survey Cake web link was left open and kept active to gather responses for approximately 13 weeks. The responses were gathered between October 2021 and January 2022. Responses were obtained from 1082 subjects who reported receiving at least one dose of the approved COVID-19 vaccines. Of 1082 responses, 889 were complete responses, with a response rate of 82.2%. Participation in this research was voluntary; therefore, the subjects received no financial compensation or any other incentives, which minimized both information and selection biases. The subjects were able to withdraw from the study anytime. The subjects who did not finish the questionnaires were excluded from the analysis.

### 2.4. Instrument

The questionnaire used in present research was divided into three categories: (a) demographic data, including sex, age, comorbidities, type of COVID-19 vaccine, and the other past history; (b) patient reported outcomes of storage LUTS before and after the first dose of the COVID-19 vaccination, both assessed by the validated Chinese version of Overactive Bladder Symptom Score (OABSS) [7]; and (c) other local and systemic AEs of COVID-19 vaccines.

### 2.5. Assessment of Storage LUTS and the Diagnosis of OAB

The storage LUTS was assessed using a validated Chinese OABSS [7], which evaluates daytime urinary frequency (score, 0–2), nocturia (score, 0–3), urinary urgency (score, 0–5), and urgency urinary incontinence (UUI) (score, 0–5). The urinary AEs were defined as the deterioration of storage LUTS, which was assessed as the increased OABSS following vaccination. Participants with an urgency score of ≥2 and an OABSS of ≥3 were considered to have OAB [8]. Diagnosis of OAB was made using pre-vaccinated OABSS.

### 2.6. Statistical Analyses

Descriptive statistics were performed for presenting the demographic variables and AE prevalence using frequencies (*n*) and percentages (%). All of the comparisons were conducted between OAB and non-OAB subjects. To identify the possible risk factors behind the deterioration of storage LUTS, univariable and multivariable logistic regression analyses were conducted to assess the effects of the clinically relevant factors. All tests were two-sided. *p*-values ≤ 0.05 were considered statistically significant. Statistical analyses were conducted using a commercial statistical software (Statistical Package for the Social Sciences, version 23.0; SPSS, Inc, Armonk, NY, USA).

## 3. Results

### 3.1. Demographic Characteristics

A total of 889 subjects with complete responses were included in the final analyses. Table 1 summarizes the baseline characteristics of all surveyed subjects. The majority (*n* = 595, 66.9%) of the subjects were women, and none of them refused to disclose their sex. All subjects had not undergone urological procedures in the past three months. Moreover, 586 (65.9%) received viral-vector-based vaccines (ChAdOx1 nCoV-19, commonly known as AstraZeneca-Oxford COVID-19 vaccine), 260 (29.2%) received mRNA-based vaccines (171 recipients of BNT162b2, commonly known as Pfizer-BioNTech COVID-19 vaccine, and 89 recipients of mRNA-1273, commonly known as Moderna COVID-19 vaccine), and 43 (4.8%) received COVID-19 candidate protein subunit vaccines (MVC-COV1901, developed by Medigen Vaccine Biologics Corp.). Seventy-one (8.0%) were diagnosed with OAB based on the pre-vaccinated OABSS. Compared to the non-OAB group, OAB was significantly associated with older ages (*p* = 0.001), higher rate of cardiovascular disease (*p* = 0.001), hypertension (*p* = 0.002), diabetes mellitus (*p* = 0.007), and hyperlipidemia (*p* < 0.001). The mean scores, including OABSS-urgency, OABSS-frequency, OABSS-nocturia, OABSS-UUI, and total OABSS, were significantly higher in the OAB group than those in the non-OAB group (all *p* < 0.001).

### 3.2. Urinary-Related AEs after COVID-19 Vaccination

Table 2 summarizes the subjects’ self-reported AEs of COVID-19 vaccines. Briefly, 119 (13.4%) subjects experienced urinary-related AEs, while 49 (5.5%) sought medical help due to urinary AEs after vaccination. Compared to the non-OAB group, OAB was significantly associated with an increased risk of urinary urgency (14.08% vs. 5.38%, *p* = 0.030), urinary frequency (11.27% vs. 5.01%, *p* = 0.027), any storage LUTS (25.35% vs. 12.35%, *p* = 0.002), and seeking medical help due to urinary-related AEs (16.9% vs. 5.2%, *p* < 0.001). There were no significant differences among generalized and musculoskeletal symptoms between the two groups.

### 3.3. Quantification of Urinary AEs with the Change of OABSS

Regarding the change in scores, the OAB group exhibited significantly greater changes in OABSS-urgency (*p* = 0.003), OABSS-frequency (*p* = 0.025), and total OABSS (*p* = 0.014) after vaccination compared to those in the non-OAB group (Table 3). Furthermore, we attempted to identify the risk factors for the urinary AEs. After adjusting for sex, age, comorbidities, and vaccine type, multivariate regression revealed that the pre-vaccinated OAB status was a risk for urinary AEs (*p* = 0.003) (Table 4).

## 4. Discussion

SARS-CoV-2 has inflicted a considerable burden on global health systems, the economy, society, and mental health [9]. By the end of February 2022, 60% of the world population and 80% of the Taiwanese population has received at least one dose of a COVID-19 vaccine [10,11]. Despite these vaccines showing proven protection against serious illness, hospitalization, and death [12], some AEs are reported after vaccination [1,2,3,4,5].

Recently, only three studies have discussed the potential for urinary-related AEs [1,5,13]. However, common AEs following COVID-19 vaccination, namely, sore arm or pain around the injection site and generalized weakness or fatigue, have been widely reported and are carefully monitored [1,2,3,4,5]. Renuka et al. investigated the AEs following COVID-19 vaccination using an online questionnaire and reported that the most common urological AE was urinary urgency, with occurrence rates of 0.75% and 1.16% following BNT162b2 mRNA COVID-19 vaccine and COVID-19 mRNA-1273 vaccine, respectively [1,5]. The second most common urological AE reported by Renuka et al. was urinary frequency with rates of 0.37% and 0.69% following BNT162b2 mRNA and mRNA-1273 vaccine, respectively. Other urinary-related AEs reported in these two studies included nocturia and urinary incontinence. Zhao et al. used the data from the FDA Vaccine Adverse Event Reporting System [13], in which all the urinary AEs were reported passively, and reported 113 urological AEs, including LUTS (*n* = 34), urinary infection (*n* = 41), hematuria (*n* = 22), skin and/or soft tissue (*n* = 16), and others (*n* = 43). However, all three studies failed to evaluate the subjects’ baseline LUTS before vaccination, making it difficult to determine the true side effects following vaccination. Furthermore, these studies used yes/no questions rather than the validated urological questionnaire to evaluate the urinary AEs, which made it impossible to determine the severity of the symptoms and might not truly represent the status, especially in those who did not know the precise definition of the storage LUTS. Therefore, to perform an accurate assessment and reduce bias, we focused specifically on a detailed change in LUTS before and after vaccination by using validated urological questionnaires. To the best of our knowledge, this is the first study to evaluate the effect of COVID-19 vaccination on the deterioration of LUTS by the change in the validated urological questionnaires.

From previous reports [1,5,13], the most common urinary-related AEs were storage LUTS, namely, urinary urgency, urinary frequency, nocturia, and urinary incontinence, which are similar to the symptoms of OAB. Therefore, we used the validated OABSS to perform the assessment. The advantages of using validated urological questionnaires included (1) being able to quantify each storage LUTS into scores, (2) having an accurate definition of scoring in each question and symptom, and (3) allowing us to evaluate the change before and after vaccination to define the urinary AEs more accurately. In the present study, the urinary AEs were defined as the deterioration of any storage LUTS, namely, the increased scores of OABSS following vaccination. We found that 13.4% subjects experienced any deterioration of storage LUTS after vaccination. Although the rate seems much higher than the rate reported in previous studies, we believe the rate might be closer to the real-world data due to the fact that the OABSS was able to evaluate storage LUTS more clearly and provide more options than the yes/no questions used previously [1,5,13].

We further attempted to clarify the association between OAB and urinary AEs and to investigate the effect of OAB on the change and severity of LUTS after vaccination. In our study, the prevalence of OAB was 8.0%, which is comparable to the prevalence rate reported in Asia of 6% to 12.4% [14,15,16,17]. In addition, OAB was significantly associated with older ages and higher rates of comorbidities, including cardiovascular disease, hypertension, diabetes mellitus, and hyperlipidemia in our study, which is consistent with the risk factors previously reported in patients with OAB [18,19,20,21]. In addition, we found that up to 25.35% of patients with OAB suffered from the worsening of any storage LUTS after vaccination and that OAB was not only significantly associated with an increased risk of urinary AEs (urinary urgency, urinary frequency, and seeking medical assistance due to the deterioration of storage LUTS) but was also significantly associated with the higher degree of deterioration of storage LUTS (namely, greater changes in OABSS-urgency, OABSS-frequency, and total OABSS after vaccination). We further identified the risk factor for the development of urinary AEs and found that OAB was the only risk factor for the post-vaccinated urinary AEs. Therefore, we suggest that clinicians should be aware of the possibility of the worsening of storage LUTS after COVID-19 vaccination, especially in subjects with OAB.

Although the worsening of storage LUTS represents a remarkable number of urinary complaints after vaccination and up to 25.35% of patients with OAB experienced worsening of storage LUTS after vaccination, only 16.9% of patients with OAB had to seek medical help. The mean and standard deviation of the post-vaccinated OABSS was found to be 6.01 ± 2.18 in the OAB group (range, 3–15). According to the assessment of the severity of OAB symptoms (mild: 0–5, moderate: 6–10, severe: 11–15 in OABSS), it can be said that the severity of the post-vaccinated storage LUTS remains moderate rather than severe in most OAB subjects despite the progression of symptoms.

Although the etiology of deterioration of storage LUTS after COVID-19 vaccine currently remains unclear, the findings may be explained by the immune response generated by the vaccine. The immune-related pathway is triggered after vaccination, which is somehow reflected in the urine. A recent study has shown that some urinary proteins related to regulated exocytosis and immune response changed obviously before and after COVID-19 vaccination [22]. Urinary protein can reveal the body’s immune response 2–4 h after vaccination [22]. However, vaccination may also activate a cytokine response or Toll-like-receptor-induced inflammation response, leading to the infiltration of antigen-presenting cells such as macrophages or dendritic cells and resulting in localized and systemic inflammatory responses [23]. Thus, bladder smooth muscle and urothelium may respond to inflammatory stimuli via secreting several substances, such as nitric oxide, ATP, and acetylcholine, which may interact with the detrusor muscle to elicit intramolecular alterations that may eventually decrease the threshold of detrusor contractility and cause storage LUTS [24,25]. However, the association between the expression of urinary proteins related to the immune response and the urinary AEs remains unclear. Therefore, further studies are needed to prove a causal association between vaccination and urinary symptoms to elucidate why patients with OAB experience more urinary AEs following vaccination than others.

In addition, recent studies demonstrated that patients with COVID-19 may have de novo or worsening of pre-existing LUTS, with irritative LUTS being the main symptoms [26,27,28]. Although the pathogenesis of these symptoms has not yet been elucidated, it was considered to be multifactorial. Previous study revealed that SARS-CoV-2 uses the angiotensin-converting enzyme 2 (ACE2) as a spike protein’s host receptor to invade target cells [29]. The ACE2 expression has been demonstrated in 2.4% of urothelial cells, which possibly increases the subsequent SARS-CoV-2-related viral cystitis [30]. The urinary bladder could be affected by SARS-CoV-2 possibly via the hematogenous route, or via urine, as viral RNA has been discovered in the urine of COVID-19 patients [31], which strengthens the hypothesis that SARS-CoV-2 may affect the urinary tract, causing an increase in urinary frequency [27]. In addition, cystitis may be secondary due to local inflammation such as endotheliitis, which plays a role in COVID-19 patients [27,32]. However, the mechanism of similar LUTS after COVID-19 vaccination and COVID-19 infection is still uncertain.

This study had some limitations. First, as this was an anonymous study, all participants who received at least one dose of vaccine were asked to finish the questionnaire at one time, which may cause recall bias. However, we believe that the results are still relevant, especially in those who suffered badly from the worsening of LUTS after vaccination, in whom the AEs were notably painful. Other non-invasive and easily performed tools, such as the uroflow stop test [33], to assess LUTS may be considered in the future studies, which may provide us with more objective results. Second, selection bias remains in our study because the questionnaire was based on the social network, which may affect the selection of patients with side effects related to the vaccine. Third, the OAB was diagnosed based on the pre-vaccination OABSS reported by the participants rather than complete clinical evaluation by a physician; therefore, the subjects in our OAB group may have some confounding factors, such as urinary tract infection, and other pathologic conditions might have been accidentally included in the OAB group. However, the OABSS is widely used for the assessment of LUTS in patients and is responsive for demonstrating the changes in OAB symptoms [34]. We believe that it is still an effective tool to quickly assess OAB symptoms. Fourth, we only investigated the storage LUTS using OABSS and did not evaluate the voiding LUTS. This is because the urinary AEs reported previously mainly included storage LUTS [1,5,13], which is carefully presented in the current study using urological validated questions. Furthermore, we were concerned that by including voiding LUTS questions, such as International Prostate Symptom Score and ICIQ-Male Lower Urinary Tract Symptoms, participants may be less willing to complete the questionnaires when they have to spend more time to perform them. However, it certainly can be carried out in the future with more modifications. Finally, SARS-CoV-2 infection may affect LUTS [26,27]; however, we did not evaluate the effect of COVID-19 infection in present study. Further studies are needed to clarify the associations.

## 5. Conclusions

Our findings demonstrated that after COVID-19 vaccination, up to 13.4% of the 889 participants revealed a deterioration in storage LUTS. Furthermore, pre-vaccination OAB was significantly associated with higher risk of worsening storage LUTS and greater change in the storage LUTS, especially urinary urgency and frequency, after COVID-19 vaccination. Although further prospective studies are needed to prove causal association between vaccine and urinary symptoms after vaccination, we suggest that the storage LUTS should be closely monitored after COVID-19 vaccination, especially in those who already suffered from OAB symptoms, and to seek for medical assistance if needed.

## Figures and Tables

**Table 1 jcm-11-02736-t001:** Baseline characteristics of all surveyed subjects.

Characteristics	All	OAB	Non-OAB	*p*-Value
Patients (*n*)	889	71	818	
**Sex**				
Male, *n*, %	294 (33.07%)	19 (26.77%)	275 (33.62%)	0.239
Female, *n*, %	595 (66.93%)	52 (73.23%)	543 (66.38%)	0.239
Age, *n*, %				0.001 *
<50 years	686 (77.2%)	44 (61.97%)	642 (78.48%)	
≥50 years	203 (22.8%)	27 (38.03%)	176 (21.52%)	
**Comorbidity**				
Cardiovascular disease	23 (2.59%)	6 (8.45%)	17 (2.08%)	0.001 *
Hypertension	75 (8.4%)	13 (18.31%)	62 (7.58%)	0.002 *
Diabetes mellitus	21 (2.4%)	5 (7.04%)	16 (1.96%)	0.007 *
Hyperlipidemia	48 (5.4%)	11 (15.49%)	37 (4.52%)	<0.001 *
Liver disease	22 (2.47%)	3 (4.23%)	19 (2.32%)	0.322
**Type of vaccine**				0.529
viral-vector-based vaccine	586 (65.9%)	45 (63.4%)	541 (66.1%)	
mRNA-based vaccine	260 (29.2%)	24 (33.8%)	236 (28.9%)	
MVC COVID-19 vaccine	43 (4.8%)	2 (2.8%)	41 (5.0%)	
**Baseline storage LUTS assessment**				
OABSS-Urgency, mean	0.40 ± 0.87	2.85 ± 1.08	0.18 ± 0.40	<0.001 *
OABSS-Frequency, mean	0.28 ± 0.50	0.79 ± 0.70	0.24 ± 0.46	<0.001 *
OABSS-Nocturia, mean	0.57 ± 0.74	1.27 ± 0.93	0.50 ± 0.69	<0.001 *
OABSS-UUI, mean	0.15 ± 0.47	0.72 ± 1.03	0.10 ± 0.34	<0.001 *
OABSS, mean	1.39 ± 1.78	5.62 ± 2.12	1.02 ± 1.15	<0.001 *

* *p* < 0.05. Abbreviations: OAB: Overactive bladder symptoms, COVID: Coronavirus disease 2019, LUTS: Lower urinary tract symptoms, OABSS: Overactive bladder symptoms score.

**Table 2 jcm-11-02736-t002:** Self-reported adverse events following COVID-19 vaccination.

Adverse Events	All *n* = 889	OAB *n* = 71	Non-OAB *n* = 818	*p*-Value
**Urinary symptoms**				
Urgency, *n* (%)	54 (6.1%)	10 (14.08%)	44 (5.38%)	0.030 *
Frequency, *n* (%)	49 (5.5%)	8 (11.27%)	41 (5.01%)	0.027 *
Nocturia, *n* (%)	69 (7.8%)	8 (11.27%)	61 (7.46%)	0.250
Urge urinary incontinence, *n* (%)	19 (2.1%)	1 (1.41%)	18 (2.20%)	0.658
Any storage LUTS, *n* (%)	119 (13.4%)	18 (25.35%)	101 (12.35%)	0.002 *
Seek medical help due to urinary-related symptoms, *n* (%)	49 (5.5%)	12 (16.90%)	37 (5.16%)	<0.001 *
**Generalized symptoms**				
Headache, *n* (%)	309 (34.8%)	24 (33.80%)	285 (34.84%)	0.860
Fever, *n* (%)	371 (41.7%)	25 (35.21%)	346 (42.30%)	0.245
Fatigue, *n* (%)	492 (55.3%)	32 (35.21%)	460 (42.30%)	0.070
Chills, *n* (%)	207 (23.3%)	13 (18.31%)	194 (23.72%)	0.301
Sore arm/injected site pain, *n* (%)	602 (37.7%)	45 (63.38%)	566 (69.19%)	0.311
**Musculoskeletal symptoms**				
Arthritis/joint pains, *n* (%)	122 (13.7%)	13 (18.31%)	109 (13.33%)	0.243
Muscle pain, *n* (%)	366 (41.2%)	29 (40.85%)	338 (41.32%)	0.938

* *p* < 0.05. Abbreviations: COVID: Coronavirus disease 2019, OAB: Overactive bladder symptoms, OABSS: Overactive bladder symptoms score, OR: Odds ratio, LUTS: Lower urinary tract symptoms.

**Table 3 jcm-11-02736-t003:** Pre- and post-vaccinated and the change in OABSS in the OAB and non-OAB groups.

Scores	Pre-Vaccinated		Post-Vaccinated		Change in Scores			
	OAB	Non-OAB	OAB	Non-OAB	OAB	Non-OAB	Total	*p*-value
	(*n* = 71)	(*n* = 818)	(*n* = 71)	(*n* = 818)	(*n* = 71)	(*n* = 818)	(*n* = 889)	
OABSS-Urgency, mean	2.85 ± 1.08	0.18 ± 0.40	2.97 ± 1.07	0.25 ± 0.56	0.13 ± 0.34	0.06 ± 0.42	0.07 ± 0.41	0.003 *
OABSS-Frequency, mean	0.79 ± 0.70	0.24 ± 0.46	0.90 ± 0.64	0.27 ± 0.48	0.11 ± 0.32	0.04 ± 0.22	0.04 ± 0.23	0.025 *
OABSS-Nocturia, mean	1.27 ± 0.93	0.50 ± 0.69	1.38 ± 0.95	0.58 ± 0.75	0.11 ± 0.32	0.08 ± 0.33	0.08 ± 0.33	0.209
OABSS-UUI, mean	0.72 ± 1.03	0.10 ± 0.34	0.76 ± 1.10	0.11 ± 0.37	0.04 ± 0.36	0.01 ± 0.18	0.01 ± 0.12	0.099
Total OABSS, mean	5.62 ± 2.12	1.02 ± 1.15	6.01 ± 2.18	1.22 ± 1.42	0.39 ± 0.80	0.19 ± 0.88	0.21 ± 0.88	0.014 *

* *p* < 0.05. Abbreviations: OABSS: Overactive bladder symptoms score, OAB: Overactive bladder, UUI: Urge urinary incontinence.

**Table 4 jcm-11-02736-t004:** Univariable and multivariable analyses of parameters for the change in OABSS after vaccination.

Variables	Change in OABSS			
	Crude OR (95% CI)	*p*-Value	Adjusted OR (95% CI)	*p*-Value
**Sex**, female	1.473 (0.953–2.276)	0.082	1.368 (0.873–2.142)	0.171
**Age**				
20–29 y	Ref.		Ref.	
30–39 y	1.087 (0.603–1.958)	0.782	1.073 (0.588–1.961)	0.818
40–49 y	1.463 (0.897–2.388)	0.128	1.402 (0.837–2.350)	0.199
50–59 y	0.590 (0.306–1.138)	0.115	0.601 (0.302–1.195)	0.147
60–69 y	1.068 (0.430–2.652)	0.888	1.169 (0.433–3.152)	0.758
≥70 y	0.549 (0.070–4.307)	0.568	0.576 (0.066–4.996)	0.617
**Comorbidity**				
Cardiovascular disease	1.375 (0.460–4.113)	0.569	1.090 (0.343–3.470)	0.884
Hypertension	0.645 (0.289–1.440)	0.285	0.706 (0.290–1.718)	0.443
Diabetes mellitus	0.318 (0.042–2.390)	0.265	0.372 (0.044–3.123)	0.362
Hyperlipidemia	0.742 (0.288–1.911)	0.536	0.915 (0.324–2.579)	0.866
**Type of vaccine**				
viral-vector-based vaccine	Ref.		Ref.	
mRNA-based vaccine	0.938 (0.610–1.442)	0.770	0.913 (0.584–1.427)	0.689
MVC COVID-19 vaccine	0.639 (0.223–1.837)	0.406	0.615 (0.211–1.791)	0.373
**Pre-vaccinated OAB**	2.411 (1.358–4.280)	0.003 *	2.552 (1.387–4.694)	0.003 *

* *p* < 0.05. Abbreviations: OABSS: Overactive bladder symptoms score, OR: Odds ratio, CI: Confidence interval.

## Data Availability

The datasets generated and/or analyzed during the current study are available from the corresponding author upon reasonable request.

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
