# Peer review of "Does COVID-19 Vaccination Cause Storage Lower Urinary Tract Symptoms?"

_jcm, 2022, doi:10.3390/jcm11102736_

Round 1

Reviewer 1 Report

In this study, the authors investigated the storage of lower urinary tract symptoms (LUTS) before and after the first dose of the coronavirus vaccine. 13.4% of patients showed a worsening of storage LUTS after vaccination.
Pre-vaccination overactive bladder (diagnosed with the Overactive Bladder Symptom Score, OABSS) was significantly associated with an increased risk of worsening urinary urgency, frequency, and seeking medical assistance. So pre-vaccinated overactive bladder was a risk for the deterioration of storage LUTS.

This study has some limitations, as stated in the text by the authors themselves. In particular, the diagnosis of the overactive bladder was evaluated exclusively through the OABSS and was evaluated only on the storage LUTS.

However, this study represents an original paper of up-to-date interest.

The manuscript is well structured and it could be suitable for publication after some minor revisions

- Why was only OABSS used to evaluate LUTS and no other questionnaires were considered? (example: IPSS and ICIQ-MLUTS). Provide an explanation in the text

- Did you include also patients who underwent a urological procedures that could impact LUTS symptoms?  

- New non-invasive tests used to study lower urinary tract symptoms, in particular urinary incontinence, have been described in the literature (doi: 10.1007/s11255-019-02107-3). It would be useful in the future to
evaluate lower urinary tract symptoms before and after the coronavirus vaccine also through these tools. It would be interesting to discuss in the text the application of these new tests

- Is it possible to stratify the results according to the type of vaccine performed? If not, explain why it was decided not to stratify the results according to the type of vaccination carried out.

- In the literature, there are studies that have evaluated LUTS before and after coronavirus infection (doi: 10.1111/ijcp.13850). Please deepen the discussion of the concept.

Author Response

Response to Reviewer 1 Comments

Comments and Suggestions for Authors: In this study, the authors investigated the storage of lower urinary tract symptoms (LUTS) before and after the first dose of the coronavirus vaccine. 13.4% of patients showed a worsening of storage LUTS after vaccination. Pre-vaccination overactive bladder (diagnosed with the Overactive Bladder Symptom Score, OABSS) was significantly associated with an increased risk of worsening urinary urgency, frequency, and seeking medical assistance. So pre-vaccinated overactive bladder was a risk for the deterioration of storage LUTS. This study has some limitations, as stated in the text by the authors themselves. In particular, the diagnosis of the overactive bladder was evaluated exclusively through the OABSS and was evaluated only on the storage LUTS. However, this study represents an original paper of up-to-date interest. The manuscript is well structured and it could be suitable for publication after some minor revisions

Point 1: Why was only OABSS used to evaluate LUTS and no other questionnaires were considered? (example: IPSS and ICIQ-MLUTS). Provide an explanation in the text.

Response 1: Thank you for the suggestion. This is indeed our limitation that we only evaluated the storage LUTS instead of voiding LUTS. Acording to previous research, the majority of people who were affected by COVID-19 vaccination suffered from storage LUTS. As a result, the aim of our study was to concentrate on storage LUTS for further analysis and discussion. Both IPSS and ICIQ-MLUTS include voiding symptoms assessments and are primarily used in men. We are concerned that the more time participants spend to complete the questionnaires, the less likely they are to complete the questionnaires. However, this is indeed a research that can be carried out in the future. We have explained it more in the section of limitation in the revised manuscript. Thank you very much.

Point 2: Did you include also patients who underwent a urological procedures that could impact LUTS symptoms?

Response 2: Thank you for the question. We apologize for not writing clearly in our article. Our questionnaire included comorbidities and any previous surgeries or procedures that subjects underwent. None of the subjects underwent urological procedures in recent 3 months. We have clarified it in the section of the results in our revised manuscript. Thank you very much.

Point 3: New non-invasive tests used to study lower urinary tract symptoms, in particular urinary incontinence, have been described in the literature (doi: 10.1007/s11255-019-02107-3). It would be useful in the future to evaluate lower urinary tract symptoms before and after the coronavirus vaccine also through these tools. It would be interesting to discuss in the text the application of these new test

Response 3: Thank you for the suggestion. This will provide us with a novel approach to future research. Instead of using only subjective questionnaires, using non-invasive tests as evaulation, such as uroflow stop test [1], will certainly provide a more in-depth point of view. We have disccussed it more in the revised manusciprt.

Reference 1: Boni, A.; Cochetti, G.; Del Zingaro, M.; Paladini, A.; Turco, M.; Rossi de Vermandois, J.A.; Mearini, E. Uroflow stop test with electromyography: a novel index of urinary continence recovery after RARP. Int Urol Nephrol. 2019, 51(4), 609-615. https://doi.org/10.1007/s11255-019-02107-3

Point 4: Is it possible to stratify the results according to the type of vaccine performed? If not, explain why it was decided not to stratify the results according to the type of vaccination carried out.

Response 4: Thank you for the great point. Yes, it is possible to stratify the results according to the different types of vaccine. In table 4 (Univariable and multivariable analyses of parameters for the change in OABSS after vaccination), we attempted to identify the risk factors for the urinary adverse events. The findings revealed that there was no statistically significant difference between different types of COVID-19 vaccines and the changes in OABSS following vaccination. Thank you very much.

Point 5: In the literature, there are studies that have evaluated LUTS before and after coronavirus infection (doi: 10.1111/ijcp.13850). Please deepen the discussion of the concept.

Response 5: Thank you very much for the great suggestion. We have added a paragraph to provide more information about the De novo or worsening of pre‐existing LUTS with COVID-19 infection in the section of discussion in the revised manuscript. Thank you very much.

Reviewer 2 Report

Good article about consequences in LUTs due to COVID19 vaccine. I recommend including in limitations that the questionnaire was based on the social network, which may affect the selection of patients with side effects related to the vaccine.

Moreover, information about LUTs worsening with COVID19 infection is also recommended

Author Response

Response to Reviewer 2 Comments

Comments and Suggestions for Authors: Good article about consequences in LUTs due to COVID19 vaccine.

Point 1: I recommend including in limitations that the questionnaire was based on the social network, which may affect the selection of patients with side effects related to the vaccine.

Response 1: Thank you for the suggestion. We have added “Selection bias remains in our study, because the questionnaire was based on the social network, which may affect the selection of patients with side effects related to the vaccine. “ in the revised manuscript. Thank you very much.

Point 2: Moreover, information about LUTs worsening with COVID19 infection is also recommended.

Response 2: Thank you for the great suggestion. We have added a paragraph to provide more information about the De novo or worsening of pre‐existing LUTS with COVID-19 infection in the section of discussion in the revised manuscript. Besides, our study focused on the association between LUTS and before and after the COVID-19 vaccination. We did not evaluate the the effect of COVID-19 infection, which was the limitation in present study. Further studies are needed to clarify the assocaitions. Thank you very much.
